# Mitochondrial Biomarkers in the Omics Era: A Clinical-Pathophysiological Perspective

**DOI:** 10.3390/ijms25094855

**Published:** 2024-04-29

**Authors:** Jacopo Gervasoni, Aniello Primiano, Michela Cicchinelli, Lavinia Santucci, Serenella Servidei, Andrea Urbani, Guido Primiano, Federica Iavarone

**Affiliations:** 1Fondazione Policlinico Universitario ‘Agostino Gemelli’ IRCCS, 00168 Rome, Italy; jacopo.gervasoni@policlinicogemelli.it (J.G.); aniello.primiano@policlinicogemelli.it (A.P.); lavinia.santucci@guest.policlinicogemelli.it (L.S.); serenella.servidei@unicatt.it (S.S.); guidoalessandro.primiano@policlinicogemelli.it (G.P.); 2Department of Basic Biotechnological Sciences, Intensive and Perioperative Clinics, Catholic University of Sacred Heart, 00168 Rome, Italy; michela.cicchinelli@unicatt.it; 3Dipartimento di Neuroscienze, Università Cattolica del Sacro Cuore, 00168 Rome, Italy

**Keywords:** mitochondrial diseases, LC-MS, metabolomics, proteomics, FTIR, biomarkers, personalized medicine

## Abstract

Mitochondrial diseases (MDs) affect 4300 individuals, with different ages of presentation and manifestation in any organ. How defects in mitochondria can cause such a diverse range of human diseases remains poorly understood. In recent years, several published research articles regarding the metabolic and protein profiles of these neurogenetic disorders have helped shed light on the pathogenetic mechanisms. By investigating different pathways in MDs, often with the aim of identifying disease biomarkers, it is possible to identify molecular processes underlying the disease. In this perspective, omics technologies such as proteomics and metabolomics considered in this review, can support unresolved mitochondrial questions, helping to improve outcomes for patients.

## 1. Introduction

Mitochondria are essential dynamic intracellular organelles, defined as the power stations of the eukaryotic cell because of their central role in providing energy in the form of adenosine triphosphate (ATP) through the oxidative phosphorylation (OxPhos). In addition, mitochondria are among the main players in cellular metabolism and homeostasis, hosting countless biochemical pathways, since the OxPhos system is connected with amino acid and carbohydrate metabolism through the tricarboxylic acid (TCA) cycle. Furthermore, they take part in the biosynthesis of metabolic precursors for macromolecules, such as proteins, lipids, RNA, and DNA, and contribute to many key cellular processes including calcium signaling, apoptosis, cellular stress response cell growth, and differentiation [1,2,3].

In 1962, with an extraordinary example of translational investigation based on morphological, biochemical, and clinical evidence, Rolf Luft described for the first time the association between a human disease condition and mitochondrial dysfunction [4], and, subsequently, he coined the term “Mitochondrial Medicine” [5]. Although this medical subspecialty currently focuses on mitochondrial dysfunctions in both common and rare diseases, this review will focus on mitochondrial diseases (MDs) and, specifically, on documenting the contribution of omics technologies in the identification of disease biomarkers in such neurogenetic disorders [6,7].

The “Mitochondrial disease” is an umbrella term that includes heterogeneous genetic disorders leading to a primary defect in mitochondrial OxPhos, which involves tissues that are heavily energy dependent and often manifest isolated neuromuscular symptoms or, more commonly, in association with multisystem involvement [8,9,10]. With an estimated minimum disease prevalence in adults of ~12.5 per 100,000 and ~4.7 per 100,000 in children [11,12,13], the primary mitochondrial disorders represent the most common group of inborn errors of metabolism. The distinctive extraordinary phenotypic variability, due to the potential involvement of any organ or tissue in the body, and for the dual genetic control with nuclear genome (nDNA) and mitochondrial DNA (mtDNA) working in concert, make these diseases a challenge for the clinician, and are responsible for frequent diagnostic odysseys [14]. The introduction of next-generation sequencing technologies (NGS), increasing the diagnostic yield and accelerating the discoveries of novel disease genes, has led to a real diagnostic revolution [15,16,17].

While enormous progress has been made in genetic characterization of MDs, similarly significant results have not been obtained in identifying disease biomarkers and in the deep understanding of the pathogenetic mechanisms [8,18].

In this context, the MDs associated with pathogenic m.3243A>G variant within *MT-TL1,* the most common heteroplasmic mtDNA disease genotype, represent an exemplary model of phenotypic heterogeneity characterizing these neurogenetic disorders. At the same time, this condition represents a challenge in understanding the pathogenetic mechanisms that underlie the clinical variability of MD patients. Frequently defined as the “MELAS (mitochondrial encephalomyopathy with lactic acidosis and stroke-like episodes) mutation”, only a percentage of about 15% of patients carrying the specific variant are associated with this canonical phenotype [19]. In the remaining cases, the subjects are characterized by being affected by PEO (progressive external ophthalmoplegia), MIDD (maternally inherited diabetes and deafness) syndromes, or a multitude of isolated or combined clinical manifestations (e.g., ataxia, isolated diabetes, myopathy, hearing impairment, cognitive decline, and short stature) that do not fall within a well-defined mitochondrial disorder [19,20]. Both for the intriguing genotype-phenotype characteristics and for the high prevalence, the m.3243A>G variant has been investigated in several omics studies aimed at identifying specific disease biomarkers and at the deep understanding of the disease. Finally, the availability of new molecules currently being, or which are shortly to be, tested in clinical trials has led to an urgent need to have reliable biomarkers able of assessing the clinical severity of the disease and monitoring the efficacy of pharmacological treatment [21,22].

In this review, we will discuss the contribution of omics in bridging this gap and show how, in parallel, the use of these technologies can provide a key contribution in the identification of pathophysiological mechanisms underlying mitochondrial diseases, thus opening up new scenarios of therapeutic options [23]. Figure 1 shows a workflow for the omic-approach.

## 2. Metabolomics and Proteomics in Mitochondrial Disease Biomarkers

Mitochondrial diseases represent a large collection of rare neurometabolic syndromes characterized by extremely difficult clinical management, both chronically and during acute events. This condition is due to a lack of complete understanding of the metabolic and molecular mechanisms involved in the pathogenesis, and also to the absence of reliable diagnostic and prognostic biomarkers that are able to identify the disease in its multiple clinical manifestations and monitor its progression. Although genetic testing provides secure diagnoses, heteroplasmy, and gaps in knowledge of pathological mechanisms limit genomics in offering a comprehensive spectrum of diseases and their variations in severity and progression. Additionally, the absence of validated biomarkers has made identifying new therapies a challenge [24,25].

Several techniques have developed to interrogate this complex process in multiple dimensions (DNA, RNA, proteins, and metabolites), known as “omics”. These disciplines allow to investigate the different classes of biological components (genes-genomic, proteins-proteomic, and metabolites-metabolomic) that determine the phenotype of an organism. Different analytical techniques (Fourier transform infrared spectroscopy (FT-IR), Raman spectroscopy, mass spectrometry (MS) and nuclear magnetic resonance spectroscopy (NMR) are used to identify either the metabolite patterns that are significant for the determination of the metabolic phenotype of the system under investigation, or the proteins used to determine of the alteration in the proteomic asset that can be identified as a possible marker of disease. Metabolomics and proteomic studies can be conducted with targeted, semi-targeted, and untargeted analytical approaches. Analysis is usually performed on multiple biological fluids: urine, saliva, plasma or serum, cerebrospinal fluid, cell cultures, tissue extracts, or biopsies. Through metabolic analysis, we can measure the metabolic profile, obtaining a fingerprint determined by the perturbation that is characteristic of the pathogenetic process [26].

Recently, the vibrational spectroscopy (Fourier Transform infrared (FT IR), near-infrared spectroscopy, and Raman spectroscopy), combined with chemometric analysis, has been used in metabolomic investigations on a variety of biological samples (i.e., tissue, cells, and other biological matrices); the vibrational frequencies are used to characterize specific spectral signatures of the molecular content of a compound for obtaining a sort of fingerprint of the sample under investigation.

The elective technique for proteomic studies is the high-resolution LC-MS that can identify previously unknown proteins and proteins with post-translational modifications from biological fluids or isolated mitochondria [27].

While these approaches individually can reveal specific pathophysiological aspects, the combined use of different omics could represent the approach capable of deeply understanding the phenotypic and severity variability of MDs [27]. Therefore, there is the need to develop integrative computational approaches to enable the integration of this type of data. The main challenges are to identify models that allow efficient selection of important characteristics and to analyze high-dimensional, scattered, and heterogeneous data [27,28].

Liquid Chromatography coupled with Tandem Mass Spectrometry is considered the gold standard technology for metabolomics investigations.

Targeted metabolomics measures specific grouping of biochemically and chemically selected metabolites. Analysis can be done in a semi-quantitative or quantitative way. The possibility to use isotopically labeled internal standards is one of the main advantages of this analytical tool for quantitative assays. The most widely used instrument platform for quantitative analysis is made by ultra-performance liquid chromatography coupled with a triple quadrupole (UPLC-QQQ). Usually, mass spectrometry experiments for quantitative analysis in diagnostic applications are conducted in Multiple Reaction Monitoring (MRM) mode to obtain a univocal molecule identification and an accurate quantification.

This strategy makes use of the thorough knowledge of a wide range of metabolic enzymes, their kinetics, final products, and the established biochemical pathways they contribute to [29].

The thorough examination of all quantifiable analytes in a sample, including chemical unknowns, is known as untargeted metabolomics. Untargeted metabolomics is a “discovery mode” method that is not relevant to individual samples; rather, it is based on differential comparison between groups. Owing to its all-encompassing character, untargeted metabolomics needs to be combined with sophisticated biostatistical methods, including multivariate analysis, in order to condense the large datasets produced into a more comprehensible set of signals. The use of silico libraries is an essential step for the discovery of new targets.

The most widely used instrument platform for untargeted analysis is made by ultra-performance liquid chromatography coupled with a high-resolution mass spectrometer (UPLC-HRMS), such as a fusion-orbitrap analyzer [30].

### 2.1. Contribution to Identifying Clinical Biomarkers

In recent years, proteomics and metabolomics have played a pivotal role in the identification of possible biomarkers, also providing a significant contribution to the physopathogenesis of mitochondrial diseases. In this section, we will focus on the contribution of the identification of potential biomarkers, underling the contribution of the studies related to the m.3243A>G variant.

Sharma et al. [24] used targeted and untargeted plasma metabolomics and proteomics measurements to identify circulating markers in a cohort of 102 patients carrying the m.3243A>G variant. The proteomic approach revealed three novel protein markers (sE-selectin, sulfate 6-O-sulfotransferase 1, and receptor tyrosine kinase) and confirmed the strong correlation of GDF15 with the severity of the disease. They used four complementary methods for metabolomic analysis and revealed N-lactoyl-amino acids, Beta-Hydroxy Fatty Acids BOHFAs, and Beta- Hydroxy Acylcarnitines BOHCAs as new families of m.3243A>G markers. This study revealed that the new biomarkers identified are associated with disease severity and to NADH-reductive stress. Proteomics validated GDF15 and nominated sE-selectin, HS6ST1, and RET as candidate biomarkers. To identify protein markers in different clinical phenotypes, they performed proteomic profiling on plasma samples from 16 m.3243A>G patients with mitochondrial encephalopathy, lactic acidosis, and stroke-like episodes (MELAS), 60 m.3243A>G non-MELAS patients (without stroke-like) [24].

In a commentary published by Gucek and Sack [27] based on the article of Sharma et al. [24], it was also underlined how difficult the search for protein biomarkers in plasma is due to the presence of dominating abundant proteins, masking the possibility to identify less abundant ones. They suggest using the affinity-based platforms as SOMAscan (an aptamer-based assay using short oligonucleotides with single-protein binding affinity) or the Olink protein platform (oligonucleotide-labeled antibodies) to measure a thousand proteins in serum with no preventive depletion. These systems allow for the determination of only predetermined targets and have low affinity for posttranslational modifications or polymorphisms. In the study, the SOMAscan platform was used to assert plasma proteomics, and four proteins met the FDR threshold that were distinct between controls and MELAS patients. The Sharma et al. study takes on a pioneering role in mitochondrial disease research, employing advanced proteomic and metabolomic techniques to investigate crucial research objectives in this field, characterizing pathological mechanisms, assessing therapeutic efficacy, and identifying potential biomarkers [24].

Esterhuizen et al. [31] examined the urine metabolomic profiles of 57 patients (9-MELAS; 30-maternally Inherited Diabetes and Deafness (MIDD); 18-mitochondrial myopathy) and healthy controls. These authors performed metabolomics analysis using different technologies (GC–TOF–MS, LC–MS/MS, and Nuclear Magnetic Resonance NMR). The data demonstrate that glucose metabolism is highly disturbed in the MIDD phenotype, while altered fatty acid synthesis is characteristic of the MELAS patients. All three phenotypes exhibited changes in levels of glycolic acid, 4-pentenoic acid, and 2-hydroxyglutaric acid. Additionally, the results led the authors to speculate about the impairment of other pathways in all three phenotypes, including the branched-chain amino acid (BCAA) metabolism.

Gervasoni et al. [32] applied FTIR spectroscopic technique on primary mitochondrial myopathies (Progressive external ophthalmoplegia [PEO]; Primary Mitochondrial Myopathy [PMM]; Oculopharyngeal muscular dystrophy [OPMD]) in the analysis of skeletal muscle from diagnostic biopsies, with the aim of identifying specific disease metabolic profiles. The results of this study revealed that the biochemical profile obtained using this technique was a sensitive and specific diagnostic marker for PEO. In particular, multivariate analysis (PLS-DA) confirms the high sensitivity and specificity of FTIR spectroscopy to stratify PEO participants with a single mtDNA deletion from those with pathogenic variants in POLG. Alix et al. [33] performed Raman spectroscopy analysis on ex vivo human muscle samples from 44 patients divided into three groups (14 mitochondrial myopathy; 13 non-mitochondrial myopathy; 17 no muscle disease). They investigated the significant differences between the groups with multivariate analyses (PCA-LDA) and revealed an overall reduction in α-helical protein content in the muscle disease group versus the no-muscle-disease group. Moreover, multivariate analysis documented a good classification accuracy between mitochondrial myopathy patients and those with non-mitochondrial myopathy. The advantages of these techniques can be identified as a fast analysis, a non-destructive approach, and an analysis of small amounts of samples required. The limitations of both studies can be identified in a small number of samples due to the rarity of the diseases.

Bocca et al. [34], in a recent study, focalized the investigations on Leber’s hereditary optic neuropathy (LHON), a frequently primary mitochondrial disorder associated with complex I deficiency, carrying out a non-targeted metabolomic investigation. They used the plasma of 18 LHON patients during the chronic phase of the disease, comparing them with healthy controls. A total of 500 metabolites were screened, of which 156 were accurately detected. The model obtained with the data analysis was able to find 13 discriminating metabolites related to the diet (nicotinamide, taurine, choline, 1-methylhistidine, and hippurate), mitochondrial energetic substrates (acetoacetate, glutamate, and fumarate), and purine metabolism (inosine). The authors interpreted the decreased concentration of taurine and nicotinamide as being interesting therapeutic targets due to their neuroprotective roles, as already documented for retinal ganglion cells [34].

With an alternative approach, Ruiz et al. [35] used comprehensive untargeted and targeted lipidomics in a case–control cohort of patients with Leigh syndrome French-Canadian variant (LSFC), a mitochondrial disease caused by variants in LRPPRC gene, and in mice with liver-specific inactivation of Lrpprc (H-Lrpprc_−/−_) unveiling a major lipid dyshomeostasis. Results from this study revealed perturbations in fatty acid (FA) metabolism in mitochondria. Higher circulating levels of various acylcarnitines (ACs), especially saturated and unsaturated long-chain acylcarnitines (LCACs), which are recognized proxies of mitochondrial FA β-oxidation. This is supported by their findings of changes in plasma and/or hepatic levels of plasmalogens, very-long-chain acylcarnitines (VLCACs), and primary bile acids conjugates, which were documented under different nutritional conditions, namely in the fasted and/or absorptive state for LSFC patients and in the fed state for mice.

Thompson Legault et al. [36] applied the GC/LC-MS approach for use in the identification of metabolomics signature using blood and urine of patients with LSFC. This study identified 45 analytes that distinguished patients from controls. Redox status, b-hydroxybutyrate/acetoacetate ratio, acylcarnitines, citric acid cycle reaction, and lipid and amino acid metabolism were altered in LSFC. These results were also validated using the principal component analysis (PCA) and permutation test, demonstrating a good separation between patients to controls.

Pastore et al. [37] analyzed other metabolites and, in particular, the balance between oxidized and reduced glutathione in lymphocytes of 10 children’s patients with genetically confirmed Leigh syndrome (LS) and 20 healthy subjects. They monitored the effects of glutathione status following 6 months of treatment with EPI-743, a para-benzoquinone analog [38]. The study included the following assessments: total, reduced, oxidized, and protein-bound glutathione levels; erythrocyte superoxide dismutase and glutathione peroxidase enzyme activities; plasma total thiols, carbonyl contents, and malondialdehyde. The authors documented a decrease in the total and reduced glutathione (GSH) associated with high levels of all oxidized glutathione forms, and a decrease in the glutathione peroxidase activity in patients. They defined glutathione as a “redox blood signature” in primary mitochondrial disorders, even if it may not be considered as the exclusive pathogenic factor responsible for clinical progression and neurodegeneration.

Buzkova et al. [39] focused their attention on a disease-specific metabolomic fingerprints detectable in blood of MDs, the inclusion of body myositis with secondary mitochondrial defects, and a mixed group characterized by a severe primary muscle dystrophies/atrophy. In particular, they analyzed the metabolomes of 25 mitochondrial myopathy or ataxias patients, 16 unaffected carriers, 6 inclusion body myositis (IBM), and 15 non-mitochondrial neuromuscular disease (NMD) patients matched with 30 controls. Serum/plasma and muscle metabolites were extracted and analyzed with triple quadrupole mass spectrometer. As a result, they showed that MD and IBM metabolomes clustered separately from controls and NMDs. MDs and IBM showed trans-sulfuration pathway changes; creatine and niacinamide depletion marked NMDs, IBM, and infantile-onset spinocerebellar ataxia (IOSCA). Low blood and muscle arginine was specific for patients with m.3243A>G mutation. They also reported that a group of four-metabolite blood biomarker (sorbitol, alanine, myoinositol, cystathionine) could distinguish primary MDs from others.

Delaney and colleagues [40] investigated the biochemical pathophysiology of two distinct forms of genetic muscle disorders. They focused the investigations on patients with McArdle disease, with a compromised ability to mobilize glucose from glycogen and consequently substrate-limited oxidative metabolism, and mitochondrial myopathy patients, comparing the obtained results with controls. The patient samples were analyzed both at rest and after exercise, highlighting that lactate and pyruvate levels at rest were lowest in McArdle patients, intermediate in controls, and highest in MD patients, and that exercise amplified these differences. Lactate and pyruvate in MD patients and controls increased comparably with exercise. On the contrary, in McArdle patients the metabolite levels were lower at baseline and remained unchanged with exercise. Furthermore, they reported in McArdle patients an elevated AMP degradation with exercise and higher resting levels of AMP degradation products, including hypoxanthine, xanthine, and inosine.

Shaham and collaborators, in a paper published in 2010 [41], explored the molecular biomarkers in cellular models and in patients affected by respiratory chain diseases (RCD). At first, they induced RC dysfunction in differentiated myotubes using rotenone and antimycin, inhibitors of NADH dehydrogenase (complex I) and ubiquinol cytochrome C oxidoreductase (complex III), respectively, to obtain extracellular metabolic profiles of the RCD starting from a simplified cellular model. Subsequently, they measured the cell culture-defined metabolites on plasma from 16 patients and 25 healthy controls. The cohort included four subjects with MELAS associated with m.3243A>G variant, one patient with the myoclonus, epilepsy, and ragged-red fibers mutation (MERRFand m.8344A>G variant), two patients with mtDNA deletions, and one patient with mtDNA depletion. They were able to detect 68 metabolites in plasma, including 26 of the 32 metabolites altered in culture media. They found that lactate and alanine were both elevated in patients. At the same time, creatine and uridine were significantly altered in patients compared to controls.

### 2.2. Contribution in Unraveling Pathophysiological Mechanisms of Disease

The increasing use of metabolic and proteomics in the identification studies of mitochondrial disease biomarkers has highlighted the central role of such omics technologies in shedding light on the underlying pathophysiological mechanisms. In this context, the metabolomic characterization of subjects with mitochondrial diseases and the m.3243A>G variant by Sharma and collaborators has brought to light an intriguing condition of metabolic adaptation focused on NADH redox imbalance characterized by elevated NADH/NAD+ [24]. This condition, defined as NADH-reductive stress, is characterized by the accumulation of substrates related to LDH-related reactions, fatty acid β-oxidation, and the TCA cycle, with possible implications on the disease manifestations. Furthermore, the overall data from this study, including the contribution of proteomics in confirming elevated GDF15 values in primitive mitochondrial disorders and in addition to previously published evidence, allowed the authors to hypothesize that NADH-reductive stress is the proximal biochemical defect leading to the activation of the integrated stress response. Focusing on a specific pathway through the use of a comprehensive untargeted and targeted lipidomics, Ruiz and colleagues revealed the characteristic metabolic perturbations in a homogeneous group of patients with mitochondrial diseases (Leigh syndrome French-Canadian variant [LSFC] associated with pathological variants in LRPPRC) and in mice harboring liver-specific inactivation of Lrpprc (H-Lrpprc^−/−^) [35]. The use of this omics technique was based on data from a previously published article coordinated by the same research group, documenting high circulating levels of long chain acylcarnitines in this specific mitochondrial disorder [36]. The results globally documented a major lipid dyshomeostasis characterized by plasma and hepatic changes in plasmalogens, bile acids conjugates, as well as very-long-chain and odd-numbered carbon chain fatty acids. This metabolic signature reflected, beyond the mitochondrion involvement, the presence of repercussions on other cellular compartments, such as the peroxisome, and suggested a functional crosstalk between these two organelles with the purpose of maintaining normal systemic and hepatic lipid metabolism. An omic approach was also carried out by Esterhuizen colleagues to define different metabolic phenotypes of subjects carrying the m.3243A>G variant. As previously mentioned, the main results of the study revealed impairment in various pathways. When characterizing the three phenotypes—MELAS, MIDD, and myopathy—some of these pathways showed consistent alterations across all, while others exhibited metabolic variations only in some, or even just in one, phenotype, revealing alterations shared among the various disease states, but also distinct variations specific to each phenotype [31].

Hall et al. conducted a study to investigate renal involvement in diabetic and non-diabetic 117 adult patients with genetically confirmed mitochondrial diseases [42]. In total, 64% (75/117) of patients carried the m.3243A>G variant, and the most frequent clinical phenotypes in the patient cohort were MIDD, MELAS, and MERRF. The research employed proteomic and metabolomic analyses of urine samples to explore potential causes of kidney abnormalities and revealed that diabetes, frequently associated with mitochondrial disorders, might not be the only cause. The metabolomics analysis showed variations in the urinary metabonomes of mitochondrial patients compared to healthy controls, observing a similar pattern in both MIDD and MELAS cases.

Furthermore, low urine levels of some metabolites have been found in MELAS patients, with respect to controls, such as N-methylnicotinamide, hippurate, and creatinine.

The study also found differences in the urinary proteome of mitochondrial patients compared to controls. In particular, the authors reported there was a decrease in the abundance of uromodulin, collagen fragments, apolipoprotein E, megalin, and cubilin. These proteins have been previously associated with decreased urinary excretion in conditions like diabetic nephropathy and other kidney diseases. Furthermore, a significant reduction in the abundance of lysosomal proteins was observed, suggesting that a mitochondrial insult in the proximal tubule could lead to abnormal functioning of the endosomal/lysosomal system.

D’Aurelio coordinated one of the most recent omics-centered investigation in this subgroup of neurometabolic disorders [43]. The authors combined gene expression and LC–MS/MS metabolomics analysis of plasma and muscle samples to delineate metabolic remodeling in a genetically and phenotypically homogeneous cohort of MDs (MERRF patients with m.8344A>G variant). In particular, the metabolic adaptations represented the result of a process consisting of distinct phases and which are part of an inter-organ crosstalk coordinated by autocrine and endocrine effects of myokines and hormones, characterized by catabolic signaling, lipid store mobilization, and intramuscular lipid accumulation.

## 3. Discussion

MDs are rare neurometabolic disorders caused by a failure of mitochondrial function and, consequently, characterized by an aberrant energy metabolism. Mitochondria are a metabolic hub, not only for the production of ATP via OxPhos, but also for the generation of biosynthetic intermediates. Playing key roles in multiple functions essential for cellular life, the dysfunction of these organelles determines the intricate metabolic and molecular repercussions that require an integrated technical approach to be correctly understood.

In the “era of personalized medicine”, characterized by the possibility of individualized care projects with treatments and medical management targeted and adapted to each patient, the need to identify specific biomarkers and a deep understanding of disease mechanisms are crucial. The aim of this review is to show how the panorama of mitochondrial diseases is very vast, and how complex it is still today to find metabolic and protein biomarkers of disease. Our intention was to highlight what has been done up to this moment and underline the urgent need to univocally identify molecules which, when dosed correctly in the clinical biochemical field, can really guide and support medical decisions, especially for the identification of personalized care paths. It would also be desirable to be able to identify biomarkers in biological fluids such as serum and urine, which are better suited to quantitative and routine analysis in patients. The continuous improvement of broadband technologies facilitates this process by transmitting detailed and integrated information concerning the physiological and pathological aspects of human conditions [44,45].

Also due to the progressive reduction of costs in the application of omics technologies, it is possible to hypothesize an increasingly wider use of these methods in the understanding of common or rare diseases (e.g., primitive mitochondrial diseases). In the future, international and interdisciplinary collaborations will be essential to develop more effective tools and share data to improve the management of complex and rare diseases, such as primary mitochondrial diseases.

## 4. Materials and Methods

To explore the existing literature on the contribution of omics technologies in the identification of biomarkers in MDs, we performed a search in the Pubmed database using the following string: (Biomarkers OR FGF21 OR GDF15 OR Gelsolin OR circulating cell-free mitochondrial DNA OR intermediary metabolism) AND (mitochondrial diseases OR mitochondrial disorders OR mitochondrial disorders OR MELAS OR mitochondrial encephalomyopathy lactic acidosis and stroke-like episodes OR MIDD OR Maternally inherited diabetes and deafness OR PEO OR progressive external ophtalmoplegia OR CPEO OR chronic progressive external ophtalmoplegia OR myoclonic epilepsy with ragged-red fibers OR MERRF OR leigh OR leber OR m.3243A>G, m.8344A>G, mtDNA deletions) AND (metabolomics OR proteomics OR transcriptomics OR omics). The last search was conducted on 1 September 2023. The search allowed to retrieve 2944 papers, of which 12 were found to be relevant to achieving the aim of this review in providing a clinical-pathophysiological perspective of disease-related pathways and biomarkers in the field of mitochondrial diseases (Table 1). Furthermore, the main results reported in the examined studies have been summarized in Table 2, where they are grouped into different categories based on the main aim of the study and classified according to the primary pathways identified.

## Figures and Tables

**Figure 1 ijms-25-04855-f001:**
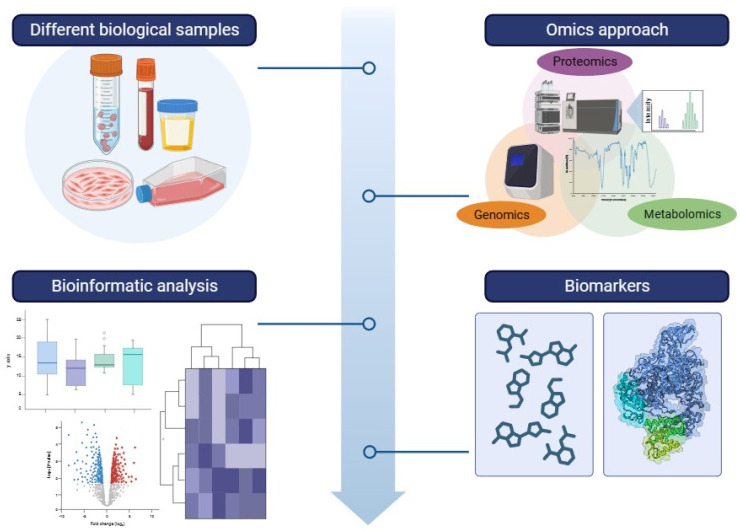
https://app.biorender.com/user/signin (accessed on 30 April 2023). A visual representation of the workflow in an omics investigation. An omics analysis starts with the collection of diverse sample types, including biological fluids (e.g., urine, saliva, blood, and cerebrospinal fluid), tissue extracts or biopsies, and cell cultures. Various analytical techniques, such as Fourier transform infrared spectroscopy (FT-IR), Raman spectroscopy, mass spectrometry (MS), and nuclear magnetic resonance spectroscopy (NMR) are employed, using targeted, semi-targeted, or untargeted approaches. The analytical results are structured into datasets and managed using bioinformatics techniques. Some identified biomolecules can emerge as potential biomarkers.

**Table 1 ijms-25-04855-t001:** In total, 12 papers resulted to be relevant to the topic of this review. No Pt: number of patients considered in the study.

Authors	Year	No Pt	Phenotype	Genotype	Techniques	Tissues Analyzed
Shaham et al. [41]	2010	16; 14	PEO, MELAS; MERRF; PMM; ME	sDel, mDel, mtDNA depletion m.3243A>G, m.8344A>G, m.8993T>G	LC-MS/MS; Proton NMR spectroscopy	media of cultured mouse muscle cell, patient plasma
Pastore et al. [37]	2013	10	LS	m.3697G>A, m.13513G>A, m.14487T>C, *SUCLA2*, *ETHE1*, *EARS2*, *SURF1*	HPLC-FL, spectrophotometric assay, colorimetric analysis, fluorometric assay	plasma; leukocytes and erythrocytes
Legault et al. [36]	2015	9	LSFC	*LRPPRC*	GC-MS and LC-MS	plasma, urine
Delaney et al. [40]	2017	21	PMM	sDel, mDel, m.3243A>G, m.10010T>C, m.12261T>C, m.4281A>G, m.8344A>G, m.5543T>C, *MT-CYB*, *ISCU*	LC-MS	plasma
Hall et al. [42]	2015	117	MELAS; MIDD; MERRF	m.3243G>A, m.8344A>G, single large scale mtDNA deletion, *POLG*, *C10orf2*/*PEO*, *COX10*, multiple mtDNA deletions	LC–MS/MS, Proton NMR spectroscopy	urine, serum
Buzkova et al. [39]	2018	26	ANS, PEO, MELAS, MIDD, IOSCA	sDel, m.3243A>G, *POLG*, *TWNK*	LC-MS/MS	serum/plasma, muscle
Ruiz et al. [35]	2019	9	LSFC	*LRPPRC*	untargeted and targeted lipidomics LC-MS (LC-QTOF and LC-QQQ)	plasma
Gervasoni et al. [32]	2020	16	PEO	sDel, *POLG*	FTIR	muscle
Sharma et al. [24]	2021	102; 16	MELAS, non-MELAS m.3243A>G	m.3243A>G	SOMAscan proteomic analysis, LC-MS	plasma
Esterhuizen et al. [31]	2021	57	MELAS, MIDD, PMM	m.3243A>G	GC-TOF-MS, LC-MS/MS, NMR	urine
Bocca et al. [34]	2021	18	LHON	m.3460G>A, m.11778G>A, m.14482C>A	untargeted metabolomics UHPLC/HRMS2	plasma
Southwell et al. [43]	2023	10	MERRF	m.8344A>G	untargeted metabolomics LC/MS, tandem LC–MS/MS	plasma/serum, muscle

**Table 2 ijms-25-04855-t002:** The table outlines the principal results extracted from the studies examined in this review, classifying them into two categories: biomarkers and pathological mechanisms, or potential new diagnostic approaches. Within the first category mentioned, the primary pathways identified as perturbed in disease states in the studies are those associated with metabolic and redox processes.

References	Proteins and Metabolites	Main Results	Field
Esterhuizen et al. [31]	In total, 54 significantly altered metabolites were discovered: 27 for the MELAS phenotype, 22 for MIDD, and 5 for myopathy.	Data highlighted an altered glucose metabolism in the MIDD phenotype, while an altered fatty acid synthesis in the MELAS patients. Hypothesized to be disturbed in all the phenotypes also the branched-chain amino acid (BCAA) metabolism.	Biomarkers and pathological mechanisms: alteration in metabolic pathways
Bocca et al. [34]	Found discriminating metabolites related to the diet (nicotinamide, taurine, choline, 1-methylhistidine and hippurate), mitochondrial energetic substrates (acetoacetate, glutamate and fumarate), and purine metabolism (inosine).	The study documents possible new therapeutic targets, in particular taurine and nicotinamide, due to their neuroprotective role.	Biomarkers and pathological mechanisms: alteration in metabolic pathways
Ruiz et al. [35]	The analysis revealed higher circulating levels of various acylcarnitines (ACs), saturated and unsaturated long-chain acylcarnitines (LCACs), changes in plasma and/or hepatic levels of plasmalogens, very-long-chain acylcarnitines (VLCACs), and primary bile acids conjugates in LSFC patients.	The study unveiled perturbations in fatty acid (FA) metabolism in mitochondria and major lipid dyshomeostasis in LSFC phenotype.	Biomarkers and pathological mechanisms: alteration in metabolic pathways
Buzkova et al. [39]	MDs and IBM showed transsulfuration pathway changes, creatine and niacinamide depletion in NMDs, IBM, and infantile-onset spinocerebellar ataxia (IOSCA). Low blood and muscle arginine in patients with m.3243A>G mutation. A group of four-metabolite (sorbitol, alanine, myoinositol, cystathionine) could distinguish primary MDs from others.	A disease-specific metabolomic fingerprint was highlighted for mitochondrial myopathy, infantile-onset spinocerebellar ataxia, inclusion body myositis, and non-mitochondrial neuromuscular diseases.	Biomarkers and pathological mechanisms: alteration in metabolic pathways
Delaney et al. [40]	In total, 9 metabolites levels differed between patient and control groups. In total, 10 metabolites levels differed between peak of exercise and rest and 17 metabolites levels differed between post-exercise and rest. The metabolites identified were associated with pyruvate metabolism, the TCA cycle, the urea cycle, and the purine nucleotide cycle.	Both at rest and exercise conditions highlight biochemical alterations that can be linked to the symptoms of the patients enrolled. In particular, lactate and pyruvate are examples of metabolites which variate across disease groups and exercise conditions.	Biomarkers and pathological mechanisms: alteration in metabolic pathways
Shaham et al. [41]	Detected 32 altered metabolites in culture media and 68 metabolites in plasma, 26 of which were had in common.	Elevated levels of lactate, alanine, creatine, and uridine were observed in patients compared with controls.	Biomarkers and pathological mechanisms: alteration in metabolic pathways
Hall et al. [42]	Metabolomics analysis highlighted low urine levels N-methylnicotinamide, hippurate, and creatinine in MDs patients respect to controls. Proteomics analysis revealed a decrease in the abundance of uromodulin, collagen fragments, apolipoprotein E, megalin, and cubilin.	The metabolomics and proteomics analysis showed variations in the urinary metabolites and proteins of mitochondrial patients compared to healthy controls.	Biomarkers and pathological mechanisms: alteration in metabolic pathways
Southwell et al. [43]	Altered TCA cycle, catabolism of glutamate, and lipid metabolism.	The study unveiled a metabolic remodeling in the cohort of MDs, characterized by catabolic signaling, lipid store mobilization and intramuscular lipid accumulation.	Biomarkers and pathological mechanisms: alteration in metabolic pathways
Legault et al. [36]	The analysis identified 45 relevant metabolites that distinguished patients from controls.	Redox status, b-hydroxybutyrate/acetoacetate ratio, acylcarnitines, citric acid cycle reaction, lipid and amino acid metabolism were altered in the disease condition.	Biomarkers and pathological mechanisms: alteration in metabolic, alteration in redox pathways
Sharma et al. [24]	Proteomic analysis revealed 3 possible novel protein markers (sE-selectin, sulfate 6-O-sulfotransferase 1, receptor tyrosine kinase). Metabolomic analysis identified N-lactoyl-amino acids, Beta-Hydroxy Fatty Acids (BOHFAs), and Beta-Hydroxy Acylcarnitines (BOHCAs) as new families of m.3243A>G markers.	The new biomarkers are associated with NADH-reductive stress, revealing an intriguing condition of metabolic adaptation focused on NADH redox imbalance that could lead to the activation of the integrated stress response.	Biomarkers and pathological mechanisms: alteration in redox pathways
Pastore et al. [37]	Glutathione forms levels and glutathione peroxidase activity were object of study.	MDs present a decrease of total and reduced glutathione (GSH), associated with high levels of all oxidized glutathione forms and a decreased of glutathione peroxidase activity. Glutathione is here defined as a “redox blood signature” and pathogenic factor in primary mitochondrial disorders.	Biomarkers and pathological mechanisms: alteration in redox pathways
Gervasoni et al. [32]	Found few peaks which can be considered putative markers for the disease discrimination.	FTIR spectroscopy emerges as a sensitive and specific diagnostic technique for PEO, revealing distinct biochemical profiles between participants with a single mtDNA deletion and those with pathogenic variants in POLG by.	New potential diagnostic approaches: spectroscopy analysis
Alix et al. [33]	Peaks assigned to proteins and various metabolites show differences in muscle disease vs. no muscle disease and between different muscle diseases.	Results demonstrated that spontaneous fiber-optic Raman spectroscopy could be applied to identify muscle disease in human samples and to discriminate between two different classes of muscle disease.	New potential diagnostic approaches: spectroscopy analysis

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
