# Peer review of "Mitochondrial Biomarkers in the Omics Era: A Clinical-Pathophysiological Perspective"

_ijms, 2024, doi:10.3390/ijms25094855_

Round 1
Reviewer 1 Report
Comments and Suggestions for Authors
This manuscript provides a review of advances gained on mitochondrial diseases' biomarkers through omics' approaches, mainly metabolomics and proteomics.
While interesting data are afforded in this review, improvement of the presentation is required and needs thorough edition.
The authors declare that only 12 published papers were found to be relevant for the issue in concern. Reporting of the findings of these 12 papers is almost the only content of the subsection 2.1. In this subsection, a revision of reported biomarkers from metabolomics and proteomics studies is basically the content. Then, a second subsection 2.2 is devoted to the altered pathways. The separation of biomarkers and pathways is somewhat confusing as one of the objectives of searching for biomarkers is to have the posibility to link them to metabolic pathways.
The following issues are suggested to be addressed and reconsidered in a new version of the manuscript:
1) Intend to group studies whose outcomes are the same metabolites or proteins, or may be related to the same metabolic pathway. If possible, explain in detail what implications or differences may be relevant for findings in different tissues or cell types. A table sumarizing the most relevant results explained in the text would improve the manuscript value.
2) Discussion is very poor. Almost only generalities are accounted for in. Try to give some conclusions in a comprehensive way and define any pattern, if any, on what is known at present and the perspectives in the near future according to the present findings.
3) The paragraphs regarding analytical techniques is poorly written (lines 83-106). LC-MS is much more important in metabolimics than vibrational spectroscopy. It does not make sense to speak about vibrational spectroscopy in much more detail than LC-MS. Indeed, in table 1 LC-MS is by far the prevailing technique.
4) Lines 77-82: this paragraph needs to be rewritten. The inteded message is difficult of being understood.
5) Lines 144-146: the same as in point 4
6) Lines 152-155: revise the message. Perhaps it is necessary to change the writing. Grammar mistakes also
7) Lines 286-288: explain better
8) Table 1: what is the meaning of "no Pt"? Explain in the table caption
Comments on the Quality of English Language
A few grammar mistakes and typing needs also revision
Author Response
We thanks the reviewer for the important suggestion for improve our manuscript, we modified the paper replying punctually to your request.
1) We added a new table in the text (table 2) were we have clearly identified the pathways differentially expressed in each work cited as relevant in table 1. In this way, the table can help in summarizing the most relevant results presented in the text.
2) We improved the discussion trying to give conclusions in a comprehensive way.
3) We corrected the paragraph adding supplemental information on LC-MS technique, and its importance in proteomic and metabolomics field for biomarker discoveries
4) The paragraph has been rewritten.
5) Also in this point, the paragraph has been rewritten.
6) The phrase has been correct and rewritten.
7)The line has been corrected.
8) The meaning of no Pt is number of patients; we added the meaning in the text.
Reviewer 2 Report
Comments and Suggestions for Authors
The authors present a narrative review of the landscape of 'omic technology/findings in mitochondrial diseases. Overall the review is well written. A few minor comments:
-perhaps provide brief, descriptions of some diseases/states that are well known to the MD community but not to those outside such as m.3243A>G variant. What is this variant and its connection to MD? why did you focus only on this specific MD? Perhaps this should be part of the intro and include the rationale for the choice.
-thank you for the table, this is helpful for summary. Are these the only 12 papers on this topic or these are the papers deemed relevant for your review?
-The "omics" use is a little mis-leading since the paper does not focus on genomic studies although you restrict your review to a genetic definition of MD.
-Have any proteomic approaches looked at protein regulation (e.g., protein phosphorylation, etc)?
Author Response
We thanks the reviewer for the important suggestion given; we modified the manuscript according to the requests.
-perhaps provide brief, descriptions of some diseases/states that are well known to the MD community but not to those outside such as m.3243A>G variant. What is this variant and its connection to MD? why did you focus only on this specific MD? Perhaps this should be part of the intro and include the rationale for the choice.
We are grateful to the Reviewer because this comment allows us to improve the manuscript and delve deeper into this important aspect. As suggested, we have added information relating to the m.3243A>G variant in the introduction.
- Pickett SJ, Grady JP, Ng YS, Gorman GS, Schaefer AM, Wilson IJ, Cordell HJ, Turnbull DM, Taylor RW, McFarland R. Phenotypic heterogeneity in m.3243A>G mitochondrial disease: The role of nuclear factors. Ann Clin Transl Neurol. 2018 Feb 7;5(3):333-345. doi: 10.1002/acn3.532.
- Nesbitt V, Pitceathly RD, Turnbull DM, Taylor RW, Sweeney MG, Mudanohwo EE, Rahman S, Hanna MG, McFarland R. The UK MRC Mitochondrial Disease Patient Cohort Study: clinical phenotypes associated with the m.3243A>G mutation--implications for diagnosis and management. J Neurol Neurosurg Psychiatry. 2013 Aug;84(8):936-8. doi: 10.1136/jnnp-2012-303528.
-thank you for the table, this is helpful for summary. Are these the only 12 papers on this topic or these are the papers deemed relevant for your review?
The 12 papers were chosen because we considered them as the most relevant for our review. We know that the literature on the argument is vast but we decided to focus the review on this 12 because the contents were fitting with our idea in presenting specific topic inside the field.
-The "omics" use is a little mis-leading since the paper does not focus on genomic studies although you restrict your review to a genetic definition of MD.
Thanks for the observation; “omics” word in this sense could be confusing. For this reason, we specified in the abstract that we restricted the field considering only proteomics and metabolomics biomarkers. The restriction for a genetic definition of MD is only a way to select the population under our attention but really, we want to show the alteration in proteomics and metabolomics pattern.
-Have any proteomic approaches looked at protein regulation (e.g., protein phosphorylation, etc)?
There are proteomic approaches that identify and analyse differences in the presence of specific PTMs, but until now, they do not provide substantial evidence that can be identified as potential biomarkers for diagnostic use in the field.
Round 2
Reviewer 1 Report
Comments and Suggestions for Authors
No comments
Comments on the Quality of English LanguageNeeds revision